# Discrimination of *Syzygium samarangense* cv. 'Giant Green' Leaves at Different Maturity Stages by FTIR and GCMS Fingerprinting

**Nuruljannah Suhaida Idris** [1], **Mohammad Moneruzzaman Khandaker** [1,*], **Zalilawati Mat Rashid** [2], **Ali Majrashi** [3], **Mekhled Mutiran Alenazi** [4], **Ahmad Faris Mohd Adnan** [5], **Khairil Mahmud** [6] **and Nashriyah Mat** [1]

1    School of Agriculture Science & Biotechnology, Faculty of Bioresources and Food Industry, Universiti Sultan Zainal Abidin, Besut Campus, Besut 22200, Terengganu, Malaysia; nuruljannahsuhaida@gmail.com (N.S.I.); nashriyahbintimat@gmail.com (N.M.)
2    School of Food Industry, Faculty of Bioresources and Food Industry, Universiti Sultan Zainal Abidin, Besut Campus, Besut 22200, Terengganu, Malaysia; zalilawati@unisza.edu.my
3    Department of Biology, College of Science, Taif University, Taif 21944, Saudi Arabia; aa.majrashi@tu.edu.sa
4    Plant Production Department, College of Food and Agricultural Sciences, King Saud University, Riyadh 11451, Saudi Arabia; amekhled@ksu.edu.sa
5    Institute of Biological Sciences, Faculty of Science, Universiti Malaya, Kuala Lumpur 50603, Selangor, Malaysia; ahmad_farisz@um.edu.my
6    Department of Crop Science, Faculty of Agriculture, Universiti Putra Malaysia, Seri Kembangan 43400, Selangor, Malaysia; khairilmahmud@upm.edu.my
*    Correspondence: moneruzzaman@unisza.edu.my; Tel.: +60-9699-3450

**Abstract:** 'Giant Green' is one of the *Syzygium samarangense* cultivars planted throughout Malaysia because it has great potential for benefitting human health. However, its variation in chemical compounds, especially in the leaves at different maturity stages, cannot be systematically discriminated. Hence, Fourier transform infrared spectroscopy (FTIR) and gas chromatography–mass spectrometry (GCMS) coupled with chemometric tools were applied to discriminate between the different stages of leaves, namely, young, mature, and old leaves. The chemical variability among the samples was evaluated by using principal component analysis (PCA) and hierarchical clustering analysis (HCA) techniques. For discrimination, partial least squares discrimination analysis (PLS-DA) was applied, and then partial least squares (PLS) was used to determine the correlation between biological activities (antioxidant and alpha-glucosidase inhibitory assay) and maturity stages of 'Giant Green' leaves. As a result, the PCA, HCA, and PLS-DA of the FTIR and GC-MS data showed the separation between clusters for the different maturity stages of the leaves. Additionally, the PLS result demonstrated that the young leaves showed a strong correlation between metabolite quantities and biological activities. The findings of this study revealed that FTIR and GC-MS coupled with chemometric analyses can be used as a rapid method for the discrimination of bioactive structural functions in relation to their biological activity.

**Keywords:** Giant Green cultivar; antioxidant; alpha-glucosidase inhibition; FTIR; GC-MS; PLS; HCA; PLS-DA; PLS



## 1. Introduction

*Syzygium samarangense*, commonly known as wax apple, jambu air, water apple, or bell fruit, is a nonclimacteric tropical fruit plant that has been cultivated in Malaysia and other neighboring countries such as Thailand, the Philippines, Vietnam, and Taiwan [1]. The three major *S. samarangense* cultivars are Giant Green, Masam Manis Pink, and Jambu Madu Red [2]. Traditionally, it is a medicinal plant: various parts are used to treat some health problems such as edema, cracked tongue, asthma, diarrhea, bronchitis, fever, ulcer, sore

throat, and to reduce blood pressure [3,4]. Additionally, the Giant Green cultivar contains an abundance of valuable phytochemicals such as phenolic acids, flavonoids, anthocyanins, and carotenes [5], which show antioxidant, antibacterial, antidiabetic, anticancer, and anti-inflammatory activities [6–8].

Metabolomics is the comprehensive analysis of a metabolite profile, either as a targeted or global application in drug discovery, phytomedicine, toxicology, and disease development [9]. Various spectroscopy and chromatography techniques are applied to detect and characterize the presence of metabolites in experimental samples [10]. Two analytical techniques for this task are Fourier transform infrared spectroscopy (FTIR) and gas chromatography–mass spectrometry (GC-MS). FTIR is an important technique used to identify the type of functional groups present in a compound. It also is a useful spectroscopic tool for profiling and fingerprinting molecular structures because it is non-destructive, simple to use, quick, and accurate [11]. The common absorption range used in plant studies is the mid-infrared (mid-IR) range. In the mid-IR range, infrared radiation is passed through a sample with a range of absorbance from 4000 $cm^{-1}$ to 400 $cm^{-1}$ [12]. Not all of the infrared radiation is absorbed by the sample: some of it passes through the sample and is transmitted to a detector. The resulting spectrum represents the molecular absorption and transmission, creating a molecular fingerprint of the sample. GC-MS is the most commonly used instrument for the separation and identification of compounds, especially in the drug discovery, pharmacology, and food industry fields [13]. The advantages of GC-MS are its low viscosity, higher sensitivity, rapid mass transfer velocity, and high resistance, so it has been widely used in chemical fingerprinting [14].

However, the abundance of metabolites present in plants poses challenges: analyzing them precisely without using a comprehensive method is difficult. Therefore, for several decades, many researchers have applied chemometric analyses coupled with spectroscopy and chromatography techniques to analyze the metabolites present in medicinal plants. Unsupervised multivariate analysis (MVDA) including principal component analysis (PCA); hierarchical clustering analysis (HCA); and supervised MVDA, including partial least squares discrimination analysis (PLS-DA) and partial least squares (PLS), are required to handle the huge dataset of the whole spectra recorded from plant samples. For example, Wijayanti et al. [15] successfully classified and discriminated the *Curcuma xanthorrhiza* from different regions using PCA and PLS-DA tools. Basyirah et al. [16] used PCA and HCA to classify and discriminate *Heterotrigona itama* propolis using different extraction methods (maceration, sonication, and Soxhlet). Additionally, PLS correlated the antioxidant activity and chemical contents of five varieties of Pegaga (Centella) extract [17]. From these studies, it can be concluded that chemometric analysis coupled with spectroscopy or chromatography techniques is a reliable tool that can be used in the metabolomics field.

Judging from the literature, it can be concluded that the metabolites in plants can be identified using spectroscopy and chromatography techniques. Additionally, the discrimination between experimental samples and the relationship between metabolites and biological activity can also be determined using multivariate data analysis (MVDA). Hence, this work aimed to discriminate the leaves of *Syzygium samarangense* cv. Giant Green at different stages of maturity and to correlate these maturity stages with their antioxidant and alpha-glucosidase inhibitory activities using FTIR- and GCMS-based metabolomics coupled with chemometrics. The findings help with identifying the most promising stages of Giant Green leaves to be used in pharmaceuticals.

## 2. Materials and Methods

### 2.1. Collection and Preparation of Plant Materials

The Giant Green cultivar of wax apple leaves, namely, young (YL), mature (ML), and old (OL) leaves, at three maturity stages were collected several times from an orchard located at Kampung Olak Lempit, Banting, Selangor, Malaysia (1028° N, 1110°20′ E), at an elevation of about 45 m above sea level. Five biological replicates of each sample were

used in this study. The leaves were selected carefully based on the below picture (Figure 1) physical examination reported by our previous study [18].

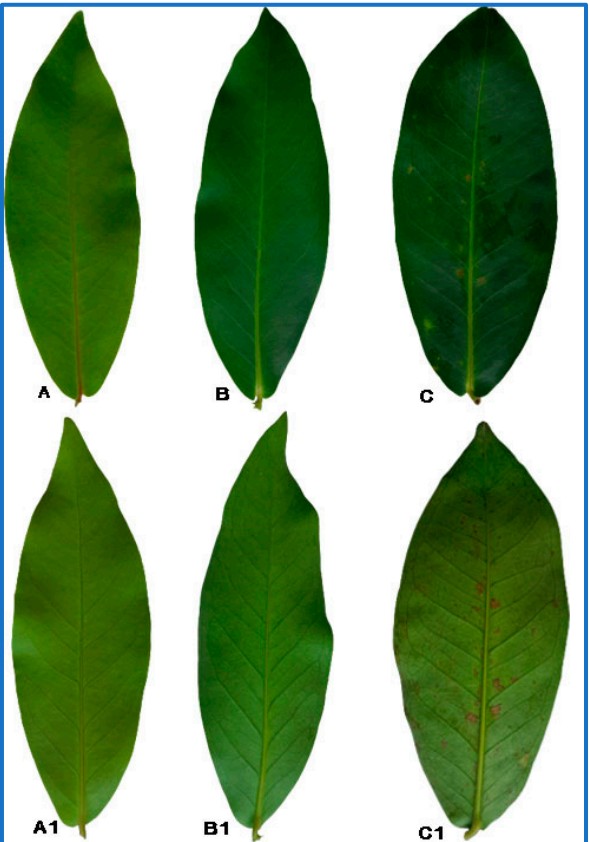

**Figure 1.** Different types of Giant Green leaves were used in this study. (**A**,**A1**)—adaxial and abaxial surfaces of the young leaf (YL), (**B**,**B1**)—adaxial and abaxial surfaces of the mature leaf (ML), and (**C**,**C1**)—adaxial and abaxial surfaces of the old leaf (OL).

*2.2. Extraction Procedure*

The fresh samples (5 g) were crushed using a mortar and pestle. The samples were soaked in methanol (25 mL) and kept for three days. Then, the extracts were heated in a water bath (70 °C for 15 min), followed by being centrifuged (1789× *g* for 15 min). The supernatants were collected and put under a fumehood until methanol was removed. The extracts were completely dried by being freeze-dried (24 h) and were stored at 4 °C before being used.

*2.3. Attenuated Total Reflectance Fourier Transform Infrared (ATR-FTIR)*

2.3.1. Sample Analysis using ATR-FTIR Machine

The diamond crystal stand was cleaned with ethanol and non-abrasive tissue. The small quantities of extracts were located on the surface stand and screwed in tightly before being analyzed by using a Shimadzu Prestige-21 Spectrophotometer (Shimadzu Brand, Kyoto, Japan) equipped with an air-cooled Deutrated Triglycin Sulphate (DTGS) detector (Shimadzu Brand, Kyoto, Japan) and scanned with a Golden Gate Single Reflection Diamond ATR accessory with an incident angle of 45° (Shimadzu Brand, Kyoto, Japan). The IR spectra of the extracts were measured with absorbance at 4000–400 cm$^{-1}$ using 4 cm$^{-1}$ and 16 scans of resolution. For each sample, three repetition measurements were collected.

2.3.2. Data Pre-Processing

The spectra were normalized and smoothed to reduce the error during data analysis. The data were saved in .txt format and then copied to Microsoft Excel.

### 2.4. Gas Chromatography-Mass Spectrometry (GC-MS)

2.4.1. Sample Preparation

The stock solution was prepared by diluting 2.5 mg of extract with 100% methanol (1 mL). For phytochemical screening, 200 μL of each leaf sample with a similar maturity stage was transferred out from the stock solution and put into the same vial for producing the final concentration of 500 μg/mL. However, for multivariate data analysis, individual samples (30 samples) were prepared by diluting 200 μL stock solution with 1 mL of 100% methanol. The solutions were vortexed for one minute and ready for analysis by using a GC-MS Agilent (19091S-433UI system) machine (Agilent Brand, Santa Clara, CA, USA).

2.4.2. GC-MS Condition

The condition system of GC-MS and Oven temperature parameters were used as below (Table 1).

**Table 1.** Condition system of GC-MS and Oven temperature parameter.

| Item | Description | | | |
|---|---|---|---|---|
| Column type | HP5MS (30 m × 250 μm) | | | |
| Film thickness | 0.25 μm | | | |
| Carrier gas | Helium | | | |
| Flow rate and pressure | 1.0 mL/min; 9.3825 Psi | | | |
| Volume of injection | 1.0 μL | | | |
| Temperature of detector | 250 °C | | | |
| Temperature of injector | 250 °C | | | |
| Temperature of oven | 80 °C | | | |
| Temperature of transfer line | 150 °C | | | |
| Mode | Splitless | | | |
| Mass scan mode | 50–55 *m/z* | | | |
| **Oven temperature parameter** | | | | |
| Item | Rate (°C/min) | Value (°C) | Hold Time (min) | Run Time (min) |
| Initial | - | 80 | 4 | 4 |
| Ramp 1 | 7 | 105 | 1 | 9 |
| Ramp 2 | 7 | 180 | 1 | 20 |
| Ramp 3 | 5 | 235 | 1 | 32 |
| Ramp 4 | 5 | 275 | 2 | 42 |

2.4.3. Data Pre-Processing

Each of the GC-MS spectra that contain the peak with a percentage of probability score of 80% and above was accepted as a particular compound and used in this analysis. For chemometric analysis purposes, the data of the percentage relative area (RA) of the compound detected in the spectrum was used. RA (%) was calculated based on the formula below [19].

Percentage of relative area = (area of particular compound/total area of all compound detected) × 100

### 2.5. Chemometric Analysis

FTIR and GC-MS spectra were subjected to four chemometric tools which are Principal Component Analysis (PCA), Hierarchical Cluster Analysis (HCA), Partial Least Squares Discriminant Analysis (PLS-DA), and Partial Least Squares (PLS). All of the data were analyzed using XLSTAT Pro 2014 software (Addinsoft, Paris, France), add-in Microsoft Excel.

2.5.1. Principal Component Analysis (PCA)

PCA is unsupervised multivariate data analysis (MVDA) which is used to find a relationship between two or more groups of 'Giant Green' leaves regarding the most variation of those variables. In the PCA technique, the new variables formed and are equal to the number of original variables. The new variables are known as principal components (PCs) and the values of new variables are known as principal component score (PCS). These

variables are not correlated with each other. The first new variable, PC1, explains the most information among the samples. Then, PC2 carries the residual information, and so on [20].

### 2.5.2. Hierarchical Cluster Analysis (HCA)

Hierarchical Cluster Analysis (HCA) is a technique combination of the same characteristic among the samples into one group or cluster. The technique of Ward's method and Euclidean distance was used for grouping 'Giant Green' leaves at different maturity stages into certain classes (clusters).

### 2.5.3. Partial Least Squares Discriminant Analysis (PLS-DA)

PLS-DA is a supervised method for classifying each sample into predefined classes. PLS-DA is complementary to PCA analysis whereas the separations between the groups of samples have been well improved. In this analysis, the dummy Y-axis (maturity stages) was responsible for separating 'Giant Green' leaves into different clusters in the score plot. The variables such as wavenumber or peak area (x-axis) that contributed to the discrimination among samples were identified from the loading plot. The global goodness of fit and quality model was confirmed by the cumulative $Q^2$, $R^2Y$, and $R^2X$ values. The accuracy and preciseness of the model were detected by using the confusion matrix. The confusion matrix represented the classifying of the observation (in percentage). The value closest to 100% shows a well-classified observation [15]. Besides, the variable importance to projections (VIP) was used to validate the variable contributed to discriminating of samples. A VIP value greater than 0.85 is known as a strong variable. The highest VIP value indicated the most relevant variable that influences the separation between the samples [21].

### 2.5.4. Partial Least Squares (PLS)

PLS is a supervised multivariate data analysis and is used when complex data with a lot of explanatory variables are involved [22]. Two variables, the dependent variable (Y) and explanatory variable (X) are used. In this study, PLS was applied to find a correlation between the FTIR fingerprint (for spectroscopy) and metabolite (for chromatography), (X) contribution in biological activities (antioxidant and alpha-glucosidase), (Y). The cumulative value of $Q^2$ is >0.5 and $R^2$ is close to 1, indicating a good model [23]. The variable (X) responsible in biological activity was identified by the variable importance in the projection (VIP). Only a VIP value greater than 0.85 indicated that a strong impact on the model was chosen.

## 3. Results

### 3.1. ATR-FTIR Fingerprint and Chemometric Analysis

3.1.1. Assignment and Comparison of ATR-FTIR Spectra

Similar patterns of IR spectra were shown in old, mature and young leaves (Figure 2). It was observed that the broad peak at 3300 cm$^{-1}$ was assigned to intermolecular hydrogen bond (O-H) of alcohol, phenol, or carboxylic acid groups [24]. Two strong signals of C=O stretching and C-N stretching were present at 1610 cm$^{-1}$ [25] and C-O stretching at 1040 cm$^{-1}$ [26]. The C-O stretching and C-C stretching was detected at 1440 cm$^{-1}$ [27], C-N stretching at 1340 cm$^{-1}$ [21] and C-O stretching or O-H bending at 1204 cm$^{-1}$ [27]. Besides, there was methylene ($CH_2$) stretching presence at 2928 cm$^{-1}$ and 2857 cm$^{-1}$ [24], C=O stretching at 1710 cm$^{-1}$ and [26,27], and C-H out-of-plane bending at 924 cm$^{-1}$ [24]. Two peaks of aromatic group presence at 824 cm$^{-1}$ and 765 cm$^{-1}$ attributed to C-H out-of-plane bending [24]. The peak at around 586 cm$^{-1}$ was assigned to the vibration of O-H out-of-plane bending [28]. The assignment of each peak is summarized in Table 2.

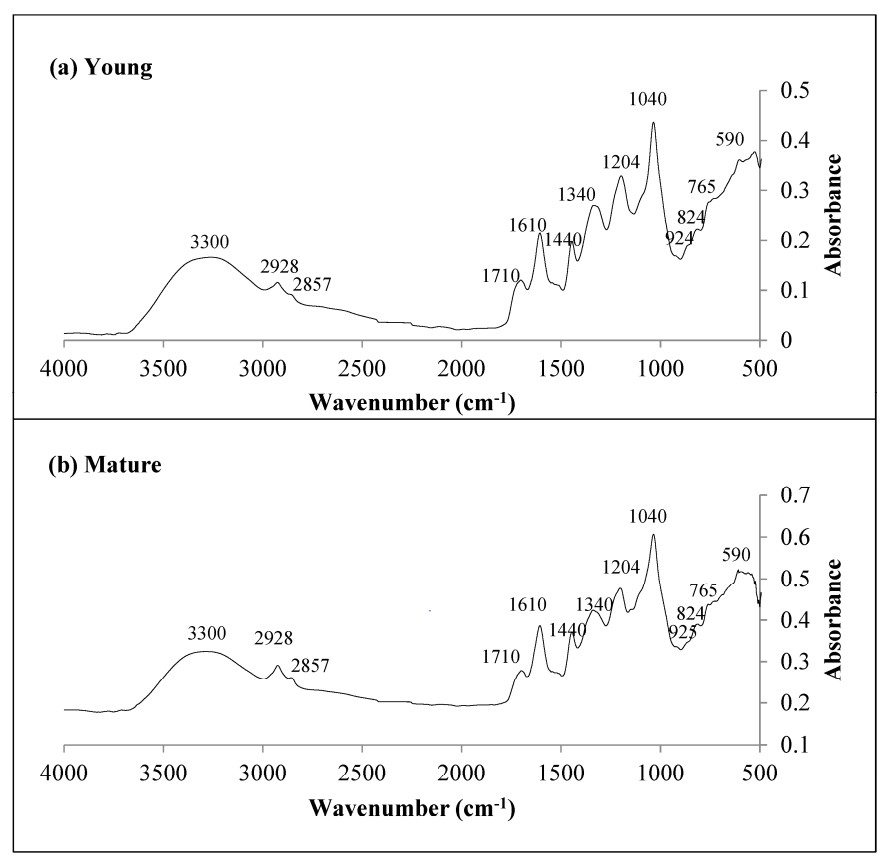

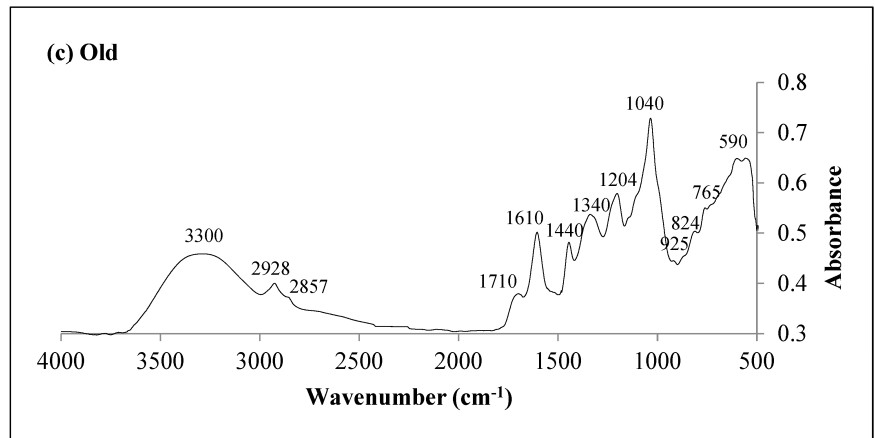

**Figure 2.** FTIR Spectra of 'Giant Green' cultivar of *S. samarangense* leaves at three maturity stages (**a**) Young leaves (**b**) Mature leaves (**c**) Old leaves.

**Table 2.** List of assignments of FTIR for 'Giant Green' cultivar of *S. samarangense* leaves at three maturity stages.

| Frequency Range (cm$^{-1}$) | Assignment of FTIR | Specific Frequency (cm$^{-1}$) | Leaves | | |
|---|---|---|---|---|---|
| | | | YL | ML | OL |
| 3500–3200 | O-H stretching | 3300 | P | P | P |
| 2942–2904 | C-H stretching asymmetric | 2928, 2933 | P | P | P |
| 2863–2846 | C-H stretching symmetric | 2857 | P | P | P |
| 1715–1710 | C=O stretching | 1710, 1715 | P | P | P |
| 1700–1600 | C=O stretching and C-N stretching (amide I) | 1610, 1642 | P | P | P |
| 1580–1510 | N-H bending and C-N stretching (amide II) | 1535 | A | A | A |
| 1450–1380 | C-O stretching and C-C stretching | 1440, 1443 | P | P | P |

**Table 2.** *Cont.*

| Frequency Range (cm$^{-1}$) | Assignment of FTIR | Specific Frequency (cm$^{-1}$) | Leaves | | |
|---|---|---|---|---|---|
| | | | YL | ML | OL |
| 1365–1343 | C-N stretching | 1340, 1345 | P | P | P |
| 1270–1150 | C-O stretching or O-H bending | 1204, 1217 | P | P | P |
| 1052–1035 | C-O stretching | 1040, 1045 | P | P | P |
| 925–910 | C-H out-of-plane bending (alkene) | 925 | P | P | P |
| 826–824 | C-H out-of-plane bending of the aromatic ring (meta) | 824 | P | P | P |
| 773–743 | C-H out-of-plane bending of the aromatic ring (para) | 765, 773 | P | P | P |
| 590–586 | O-H out-of-plane bending of alcohol | 590 | P | P | P |

P = present, A = absent, YL = young leaves, ML = mature leaves, and OL = old leaves.

### 3.1.2. Chemometric Analysis

### Principal Component Analysis (PCA)

PCA was performed in this study for unsupervised classification of leaves of the 'Giant Green' cultivar of *S. samarangense* at different maturity stages. Based on the score plot (Figure 3A), the total variance accounting for the first two principal components in the leaves extract was 96.64% (PC1: 50.57%; PC2: 46.07%) (Figure 3B). The model showed a separation between the maturity stages of leaf samples. The interpretation of the score plot within the loading plot gave a clear picture of the factor influencing the clustering of leaf extracts. The loading plots with a score ≥0.75 were accepted as a strong factor. Table 3 shows the variables that contributed to leaf variation along PC1 and PC2. The loading plots of leaf extracts revealed that wavenumbers at 3300, 2928, 1710, 1610, 1440, 1340, 1204, 1040, 924, 824, 765, and 590 cm$^{-1}$ contributed to variation in PC1.

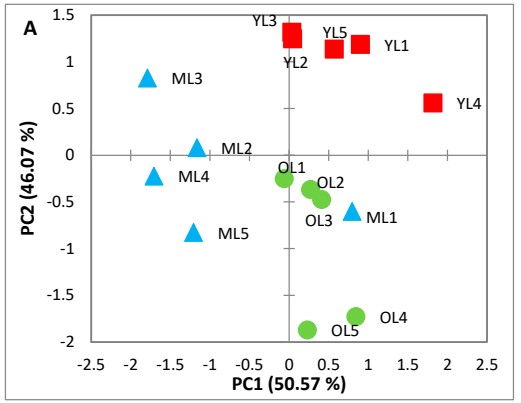 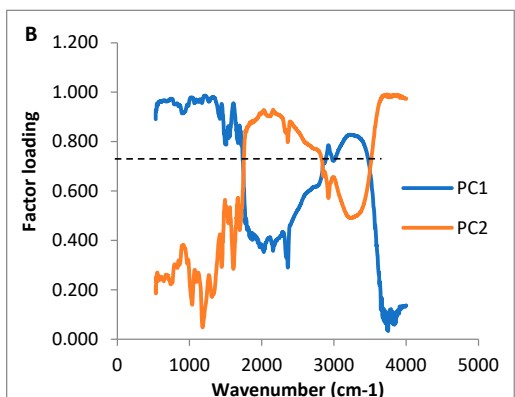

**Figure 3.** PCA-Derived of FTIR spectra representing 'Giant Green' cultivar of *S. samarangense* leaves at three maturity stages (**A**) Score Plot (**B**) Loading Plot of PC1 and PC2.

**Table 3.** Summary of strong loading variables (≥0.75) on the varimax rotation of principal component (PCs) analysis for the 'Giant Green' cultivar of *S. samarangense* leaves.

| Variable | Name of Metabolite | PCs |
|---|---|---|
| 3 | Cyclotetradecane | PC1 |
| 8 | 2,6,11,15-Tetramethylhexadecane | PC2 |
| 9 | 1-Iodododecane | PC2 |
| 10 | 9-Methyl-1-undecene | PC1 |
| 11 | 2-Butyl-1-decene | PC1 |
| 12 | (E)-9-Eicosene | PC1 |
| 14 | Phthalic acid, butyl hept-4-yl ester | PC2 |
| 21 | Phosphonofluoridic acid, methyl-, nonyl ester | PC2 |
| 23 | Diethylene glycol dibenzoate | PC1 |
| 25 | Decanol | PC1 |
| 27 | Hexadecanol | PC1 |
| 29 | Octadecanol | PC1 |
| 32 | 6,10,14-Trimethyl-2-pentadecanone | PC2 |

Hierarchical Cluster Analysis (HCA)

The similarities and differences among the three maturity stages of 'Giant Green' leaves were evaluated with HCA analysis (Figure 4). Three clusters of leaf samples were suggested. Cluster one contained all old leaf replicates and one replicate from the mature leaf sample. Then, cluster two contained the rest of the mature leaf replicates and cluster three contained all young leaf replicates. The sample formed with the same cluster tended to have a high similarity parameter.

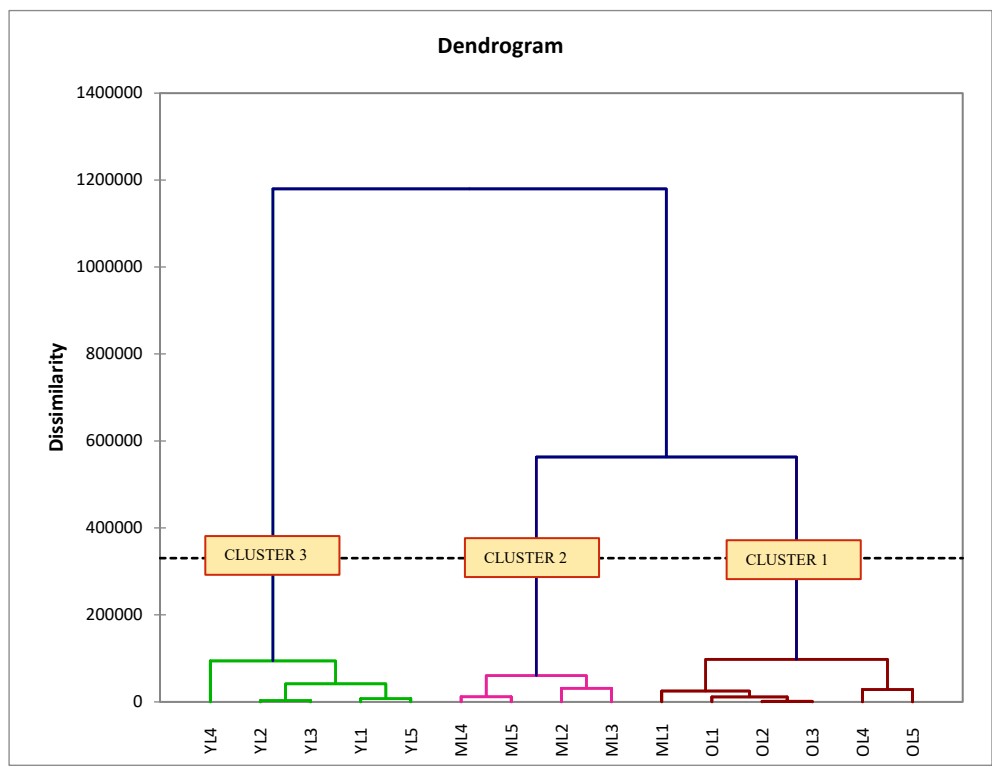

**Figure 4.** Dendrogram from Hierarchical Cluster Analysis (HCA) of 'Giant Green' cultivar of *S. samarangense* leaves at three maturity stages.

Partial Least Square–Discriminant Analysis (PLS-DA)

PLS-DA belongs to the supervised pattern recognition method. It was complementary to PCA analysis. It also had abilities to improve the separation between the groups of samples. As seen in Figure 5A,B, the separation of the three maturity stages of leaf samples was improved. This PLS-DA model had an overall $Q^2$ cumulative of 0.444, $R^2Y$ cumulative of 0.608, and $R^2X$ cumulative of 0.972. The model had a $Q^2$ cumulative value of <0.5, indicating no global goodness of fit. This suggested that the quality of the fit varies a lot depending on the maturity stage of the leaves. Besides, the efficiency of PLS-DA in classifying and discriminating the samples can be accessed through the confusion matrix. The confusion matrix result showed that all of the leaf extracts have been classified with 93.33% of correction. Besides, the variable importance to projections (VIP) was used to validate the variable contributed to the discriminating of samples. Most of the peaks have VIP values greater than 0.85 except 1204 and 1040 cm$^{-1}$ which contributed the highest in discrimination between young, mature, and old leaves. The overall VIP values are shown in Table 4.

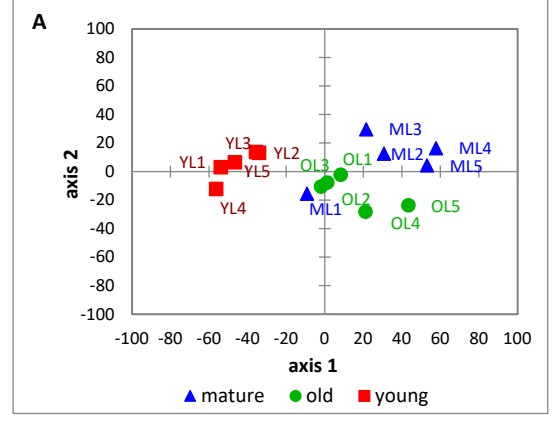
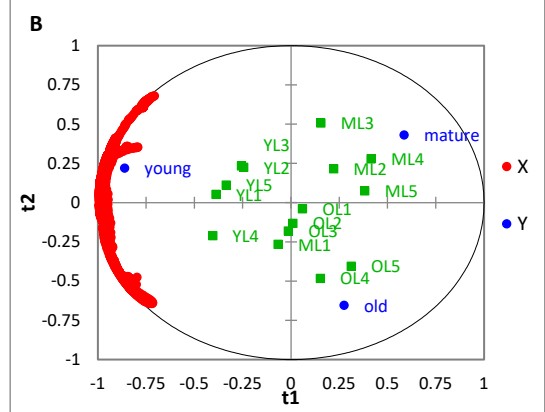
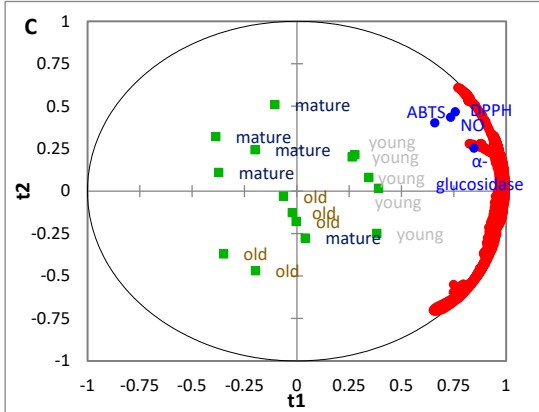

**Figure 5.** The PC1 and PC2 of FTIR results of the 'Giant Green' cultivar of *S. samarangense* leaves at three maturity stages of (**A**) PLS-DA score plot (**B**) PLS-DA bi-plot (X = FTIR wavenumber; Y = leaf maturity stages) (**C**) PLS bi-plot.

**Table 4.** Summary of strong variable importance to projections (VIP) scores (≥0.85) correspond to the partial-least square-discrimination analysis (PLS-DA) and partial-least square analysis (PLS) of 'Giant Green' cultivar of *S. samarangense* leaves at three maturity stages.

| Variables (cm⁻¹) | Assignment of FTIR (PLS-DA) | VIP Score | |
|---|---|---|---|
| 2857 | C-H stretching symmetric | 1.11 | |
| 2928 | C-H stretching asymmetric | 1.10 | |
| 1709 | C=O stretching | 1.02 | |
| 924 | C-H out-of-plane bending (alkene) | 1.00 | |
| 3300 | O-H stretching | 0.98 | |
| 1439 | C-O stretching and C-C stretching | 0.96 | |
| 824 | C-H out-of-plane bending of the aromatic ring (meta) | 0.96 | |
| 1609 | C=O stretching and C-N stretching (amide I) | 0.94 | |
| 764 | C-H out-of-plane bending of the aromatic ring (para) | 0.90 | |
| 590 | O-H out-of-plane bending of alcohol | 0.89 | |
| 1339 | C-N stretching | 0.85 | |
| **Variables (cm⁻¹)** | **Assignment of FTIR (PLS)** | **VIP Score** | **PCs** |
| 2857 | C-H stretching symmetric | 1.0 | PC1 |
| 2928 | C-H stretching asymmetric | 0.89 | PC1 |
| 3300 | O-H stretching | 0.87 | PC1 |

Partial Least Square (PLS)

PLS was established to investigate the relationship between bioactivities of antioxidants and alpha-glucosidase (Y variable) with FTIR fingerprint (x variable). The details about the antioxidant and alpha-glucosidase inhibitory activities of 'Giant Green' leaves at three maturity stages were reported in our previous study [18]. As seen in the bi-plot (Figure 5C), most young leaf samples were located at the upper right-hand quadrant on the t1 axis. This revealed that the young leaves possessed the strongest bioactivities as compared to mature and old leaves. This analysis showed a good PLS prediction model with the cumulative values of $Q^2$ at 0.591, $R^2Y$ at 0.724, and $R^2X$ at 0.972. The peaks related to this relationship were evaluated based on the loading plot (w*c). The influencer peaks were 2857, 2928, and 3300 cm$^{-1}$ which were detected to have the highest w*c [1] values. This result also aligned with data from VIP coefficients where the peak at 2857 cm$^{-1}$ had the highest VIP value than other peaks. The overall VIP values are shown in Table 4.

*3.2. Gas Chromatography-Mass Spectrometry (GC-MS) and Chemometric Analysis*

3.2.1. Assignment and Comparison of GC-MS Spectra

Based on the chromatogram shown in Figure 6a–c, the total number of metabolites identified in leaf extracts was 37. The quantities of metabolites that were found in each of the samples were not the same, whereby in young leaves it was 25 (Figure 6a), in mature leaves it was 29 (Figure 6b) and in old leaves it was 27 (Figure 6c). Six major metabolites were detected in all of the maturity stages of leaf samples which were aligned in similar retention times. These major metabolites were cyclotetradecane; octadecanol; 5(2,4-di-tert-butylphenoxy)-5-oxopentanoic acid; methoxyl; hexadecanol; and demethoxymatteucinol. Moreover, some metabolites were detected only in one or two leaf samples, while some of them were also present in all the samples but differed in peak intensities. For example, the metabolites of methyl (9Z,15Z)-9,15-octadecadienoate and stearic acid, butyl ester were only detected in young leaves but 2,6,11,15-tetramethylhexadecane, 1-iodododecane and phytol were detected in mature and old leaves. The detailed information on these metabolites is tabulated in Table 5.

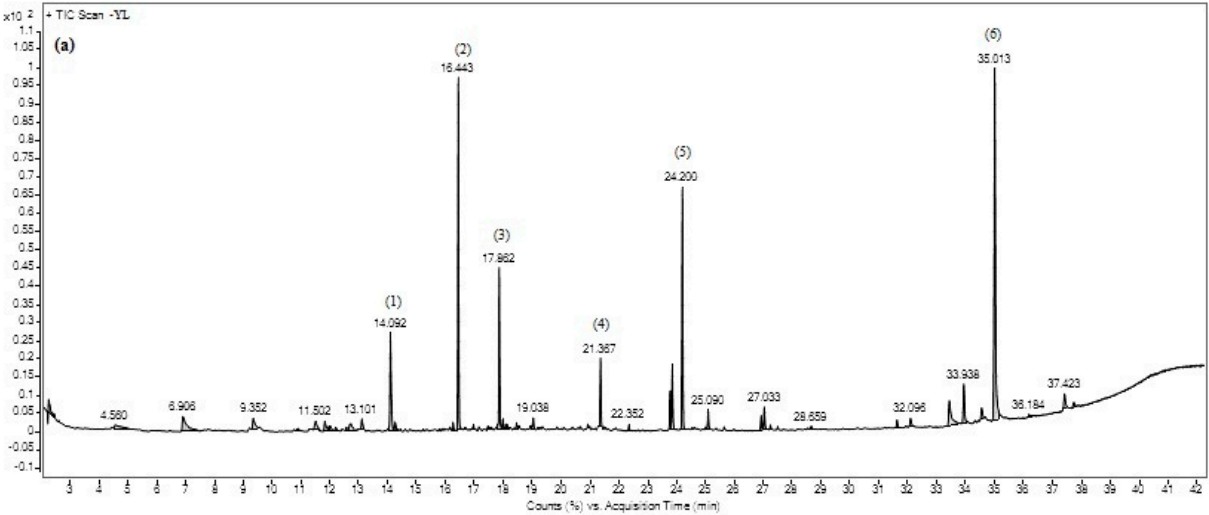

**Figure 6.** *Cont.*

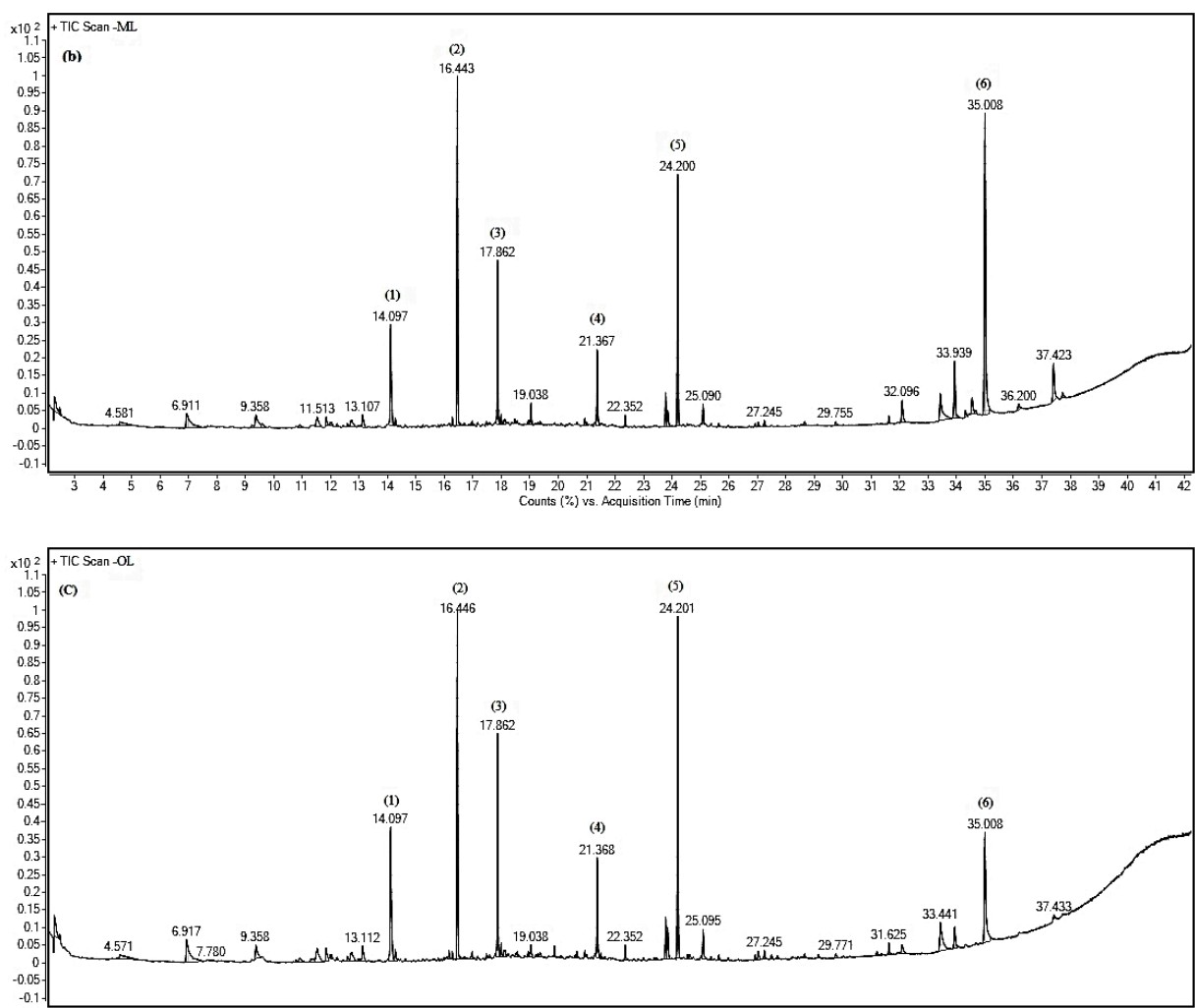

**Figure 6.** GC-MS chromatogram of major volatile metabolites present in 'Giant Green' cultivar of *S. samarangense* leaves at three maturity stages (**a**) young leaves (**b**) mature leave (**c**) old leaves; (1): cyclotetradecane; (2): 5(2,4-di-tert-butylphenoxy)-5-oxopentanoic acid; (3): hexadecanol; (4): octadecanol; (5): methoxyl; and (6): demethoxymatteucinol.

**Table 5.** List of metabolites present in the 'Giant Green' cultivar of *S. samarangense* leaves at three maturity stages with a percentage of probability score of 80% and above.

| Group | Variable | Name of Metabolite | RT (Min) | Molecular Formula | Relative Area (%) | | |
|---|---|---|---|---|---|---|---|
| | | | | | YL | ML | OL |
| Alkane | 1 | Decane | 4.57 | $C_{10}H_{22}$ | $0.94 \pm 0.16$ [a] | $0.73 \pm 0.25$ [a] | $0.39 \pm 0.53$ [a] |
| | 2 | 2,3-Dimethyloctane | 12.73 | $C_{10}H_{22}$ | $0.09 \pm 0.21$ [a] | $0.09 \pm 0.20$ [a] | $0.10 \pm 0.22$ [a] |
| | 3 | Cyclotetradecane | 14.1 | $C_{14}H_{28}$ | $5.79 \pm 0.82$ [a] | $4.78 \pm 0.54$ [a] | $4.57 \pm 1.21$ [a] |
| | 4 | Tetradecane | 14.26 | $C_{14}H_{30}$ | $0.40 \pm 0.07$ [a] | $0.26 \pm 0.15$ [ab] | $0.14 \pm 0.91$ [b] |
| | 5 | 3,5-Dimethylundecane | 16.97 | $C_{13}H_{28}$ | $0.09 \pm 0.13$ [a] | $0.05 \pm 0.11$ [a] | $0.24 \pm 0.31$ [a] |
| | 6 | 4,6-Dimethyldodecane | 16.15 | $C_{14}H_{30}$ | ND | $0.06 \pm 0.14$ [a] | ND |
| | 7 | 6-Ethyl-2-methyldecane | 17.99 | $C_{13}H_{28}$ | $0.06 \pm 0.14$ [a] | ND | ND |
| | 8 | 2,6,11,15-Tetramethylhexadecane | 19.86 | $C_{20}H_{42}$ | ND | $0.12 \pm 0.27$ [a] | $0.22 \pm 0.34$ [a] |
| | 9 | 1-Iodododecane | 20.66 | $C_{12}H_{25}I$ | ND | $0.05 \pm 0.12$ [a] | $0.05 \pm 0.11$ [a] |
| Alkene | 10 | 9-Methyl-1-undecene | 11.52 | $C_{12}H_{24}$ | $0.91 \pm 0.16$ [a] | $0.78 \pm 0.08$ [a] | $0.64 \pm 0.30$ [a] |
| | 11 | 2-Butyl-1-decene | 13.11 | $C_{14}H_{28}$ | $0.74 \pm 0.10$ [a] | $0.61 \pm 0.08$ [a] | $0.51 \pm 0.29$ [a] |
| | 12 | (E)-9-Eicosene | 25.09 | $C_{20}H_{40}$ | $0.90 \pm 0.17$ [a] | $0.79 \pm 0.09$ [a] | $0.66 \pm 0.38$ [a] |
| Ether | 13 | Decyl octyl ether | 11.83 | $C_{18}H_{38}O$ | $0.12 \pm 0.28$ [a] | $0.10 \pm 0.22$ [a] | $0.20 \pm 0.27$ [a] |
| Ester | 14 | Phthalic acid, butyl hept-4-yl ester | 24.54 | $C_{19}H_{28}O_4$ | ND | $0.03 \pm 0.07$ [a] | $0.04 \pm 0.08$ [a] |
| | 15 | Methyl benzoate | 6.92 | $C_8H_8O_2$ | $2.00 \pm 0.22$ [a] | $1.76 \pm 0.49$ [a] | $1.65 \pm 0.15$ [a] |
| | 16 | Bis(2-ethylhexyl) carbonate | 10.9 | $C_{17}H_{34}O_3$ | $0.37 \pm 0.71$ [a] | ND | $0.07 \pm 0.10$ [b] |
| | 17 | 5(2,4-Di-tert-butylphenoxy)-5-oxopentanoic acid | 16.44 | $C_{19}H_{28}O_4$ | $10.43 \pm 1.20$ [a] | $8.86 \pm 1.17$ [a] | $8.89 \pm 1.92$ [a] |
| | 18 | Methyl palmitate | 23.85 | $C_{17}H_{34}O_2$ | $2.00 \pm 1.04$ [a] | $0.49 \pm 0.30$ [b] | $0.50 \pm 0.46$ [b] |
| | 19 | Methylox | 24.20 | $C_{18}H_{28}O_3$ | $8.80 \pm 1.23$ [a] | $7.40 \pm 0.62$ [a] | $7.95 \pm 0.73$ [a] |
| | 20 | Methyl (9Z,15Z)-9,15-octadecadienoate | 26.92 | $C_{19}H_{36}O_2$ | $0.07 \pm 0.16$ [a] | ND | ND |

**Table 5.** *Cont.*

| Group | Variable | Name of Metabolite | RT (Min) | Molecular Formula | Relative Area (%) | | |
|---|---|---|---|---|---|---|---|
| | | | | | YL | ML | OL |
| | 21 | Phosphonofluoridic acid, methyl-, nonyl ester | 27.03 | $C_{12}H_{26}FO_2P$ | ND | ND | 0.04 ± 0.08 [a] |
| | 22 | Stearic acid, butyl ester | 31.95 | $C_{22}H_{44}O_2$ | 0.21 ± 0.46 [a] | ND | ND |
| | 23 | Diethylene glycol dibenzoate | 33.44 | $C_{18}H_{18}O_5$ | 2.38 ± 0.53 [a] | 2.23 ± 0.33 [a] | 1.84 ± 1.07 [a] |
| | 24 | 4-Methylhexanol | 12.02 | $C_7H_{16}O$ | 0.09 ± 0.13 [a] | 0.04 ± 0.10 [a] | ND |
| Alcohol | 25 | Decanol | 9.36 | $C_{10}H_{22}O$ | 1.35 ± 0.23 [a] | 1.13 ± 0.08 [a] | 0.99 ± 0.56 [a] |
| | 26 | Dodecanol | 14.12 | $C_{12}H_{26}O$ | 7.55 ± 6.68 [b] | 10.16 ± 6.84 [b] | 4.83 ± 5.08 [c] |
| | 27 | Hexadecanol | 17.86 | $C_{16}H_{34}O$ | 6.07 ± 0.89 [a] | 5.04 ± 0.54 [a] | 4.92 ± 1.09 [a] |
| | 28 | Intermedeol | 19.04 | $C_{15}H_{26}O$ | ND | 0.33 ± 0.32 [a] | 0.11 ± 0.24 [a] |
| | 29 | Octadecanol | 21.37 | $C_{18}H_{38}O$ | 3.06 ± 0.48 [a] | 2.61 ± 0.28 [a] | 2.47 ± 0.69 [a] |
| | 30 | Phytol | 27.25 | $C_{20}H_{40}O$ | ND | 0.15 ± 0.33 [a] | 0.05 ± 0.12 [b] |
| Ketone | 31 | 2-(1,1-Dimethylethyl)-cyclobutanone | 18.1 | $C_8H_{14}O$ | ND | ND | 0.04 ± 0.10 [a] |
| | 32 | 6,10,14-Trimethyl-2-pentadecanone | 22.35 | $C_{18}H_{36}O$ | ND | 0.13 ± 0.18 [a] | 0.23 ± 0.34 [a] |
| | 33 | 7,9-Di-tert-butyl-1-oxaspiro(4,5)deca-6,9-diene-2,8-dione | 23.77 | $C_{17}H_{24}O_3$ | 1.32 ± 0.30 [a] | 0.96 ± 0.21 [ab] | 0.69 ± 0.47 [b] |
| | 34 | Pinostrobin chalcone | 32.1 | $C_{16}H_{14}O_4$ | ND | 0.54 ± 0.53 [a] | 0.28 ± 0.38 [a] |
| | 35 | Pinocembrin | 33.94 | $C_{17}H_{16}O_4$ | 0.72 ± 0.72 [a] | 1.66 ± 0.98 [b] | 0.78 ± 0.67 [a] |
| | 36 | Demethoxymatteucinol | 35.02 | $C_{17}H_{16}O_4$ | 0.03 ± 0.07 [a] | ND | ND |
| Amine | 37 | 5-Aminotetrazole | 9.58 | $CH_3N_5$ | 1.30 ± 2.90 [a] | ND | 0.51 ± 1.14 [b] |

ND = not detected. YL = young leaves; ML = mature leaves; OL = old leaves. Values are the means ± standard deviation for five biological replicates of experiments (n = 5). Data from the same horizontal row with different superscript letters refer to a significant difference ($p < 0.05$).

### 3.2.2. Chemometric Analysis

### Principal Component Analysis (PCA)

The PCA analysis was established to give better information about the similarities and differences between the three maturity stages of leaves of 'Giant Green' in the context of their metabolites. It can be seen from Figure 7A, the score plot of leaf samples revealed that the total variance of the first two principal components was 43.50% with values of PC1 at 27.79% and PC2 at 15.71%. The model showed no separation between the maturity stages of leaf samples. The loading line plot (Figure 7B) of leaf samples identified the metabolites contributing to the variation in PC1 that were 3, 10, 11, 12, 23, 25, 27, and 29. Meanwhile, the metabolites 8, 9, 14, 21, and 32 contributed to PC2. The details about the metabolites are explained in Table 6.

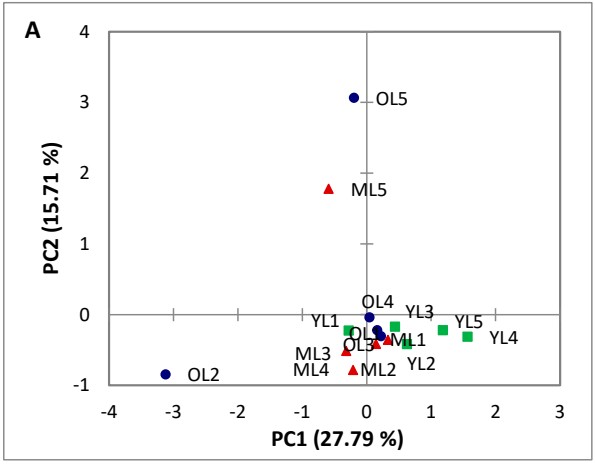
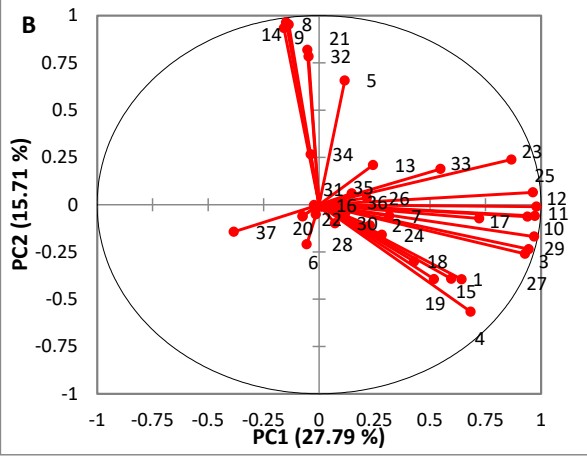

**Figure 7.** PCA-Derived of the GC-MS result representing the 'Giant Green' cultivar of *S. samarangense* leaves at three maturity stages (**A**) Score Plot (**B**) Loading Plot of PC1 and PC2.

**Table 6.** Summary of strong loading variables (≥0.75) on the varimax rotation of principal component (PCs) analysis for the 'Giant Green' cultivar of *S. samarangense* leaves.

| Variable | Name of Metabolite | PCs |
|---|---|---|
| 3 | Cyclotetradecane | PC1 |
| 8 | 2,6,11,15-Tetramethylhexadecane | PC2 |
| 9 | 1-Iodododecane | PC2 |
| 10 | 9-Methyl-1-undecene | PC1 |
| 11 | 2-Butyl-1-decene | PC1 |
| 12 | (E)-9-Eicosene | PC1 |
| 14 | Phthalic acid, butyl hept-4-yl ester | PC2 |
| 21 | Phosphonofluoridic acid, methyl-, nonyl ester | PC2 |
| 23 | Diethylene glycol dibenzoate | PC1 |
| 25 | Decanol | PC1 |
| 27 | Hexadecanol | PC1 |
| 29 | Octadecanol | PC1 |
| 32 | 6,10,14-Trimethyl-2-pentadecanone | PC2 |

Hierarchical Cluster Analysis (HCA)

The cluster analysis of 'Giant Green' leaves was illustrated clearly in the HCA dendrogram. Based on GC-MS data, the samples were grouped into three groups. The result of HCA revealed that all of the leaf samples in similar maturity stages formed a heterogeneous cluster (Figure 8). Cluster one represented three replicates from old leaves and young leaves and two replicates from mature leaves. Cluster two represented two replicates from mature and young leaves and one replicate from old leaves. Cluster three represented one replicate from mature and old leaves.

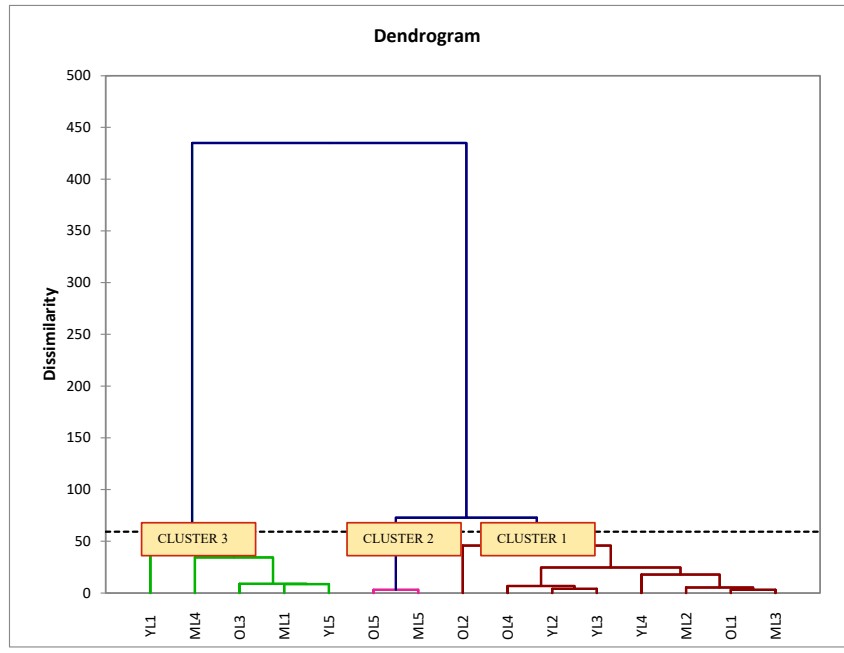

**Figure 8.** Dendrogram from Hierarchical Cluster Analysis (HCA) corresponded to GC-MS data for the 'Giant Green' cultivar of *S. samarangense* leaves at three maturity stages.

Partial Least Square-Discriminant Analysis (PLS-DA)

PLS-DA analysis was done to improve the separation between leaf samples obtained from PCA results. Pattern recognition of PLS-DA from leaves extract was carried out and is shown in Figure 9A,B. The results revealed that the separation between the maturity stages of leaf samples had improved. The model had a $Q^2$ cumulative of 0.875, $R^2Y$ cumulative of 1.000, and $R^2X$ cumulative of 0.959. This model was indicated as a good model because it

had a $Q^2$ cumulative value greater than 0.5. The result of the confusion matrix also showed that young, mature, and old leaves were discriminated efficiently with a 100% correctly produced overall classification rate with no misclassified sample. Based on the results of the variable importance to projections (VIP), 24 metabolites were identified that influenced the separation of young, mature, and old leaf samples. These metabolites were 18, 33, 4, 3, 27, 1, 19, 11, 29, 17, 15, 10, 34, 25, 32, 24, 12, 16, 20, 22, 36, 7, 8, and 28. The overall result is summarized in Table 7.

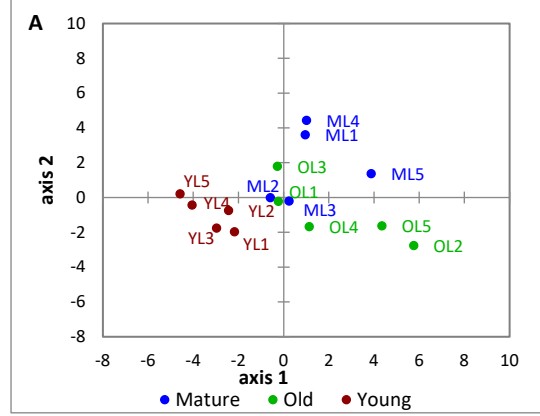
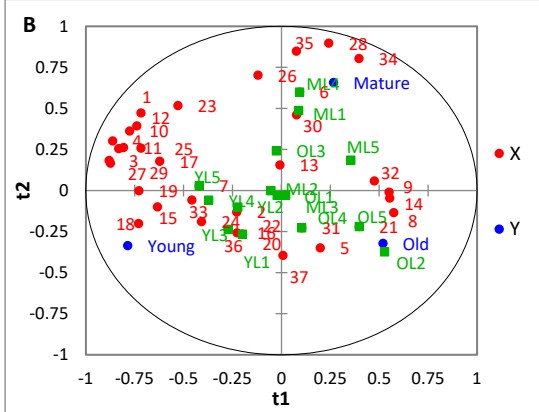

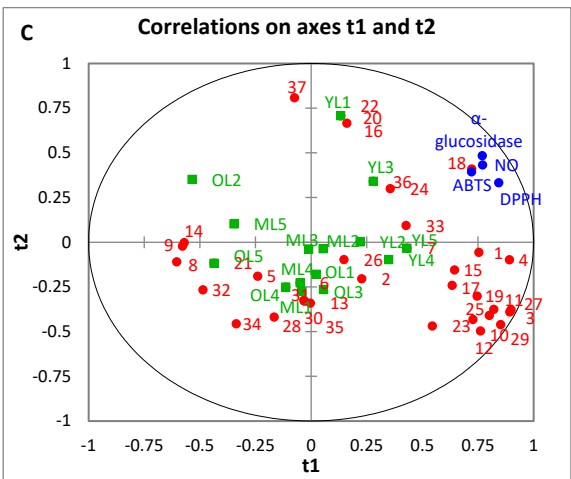

**Figure 9.** The PC1 and PC2 of GC-MS results of the 'Giant Green' cultivar of *S. samarangense* leaves at three maturity stages of (**A**) PLS-DA score plot (**B**) PLS-DA bi-plot (X = metabolites; Y = leaf maturity stages) (**C**) PLS bi-plot.

**Table 7.** Summary of strong variable importance to projections (VIP) scores ($\geq$0.85) correspond to the partial-least square-discrimination analysis (PLS-DA) of the 'Giant Green' cultivar of *S. samarangense* leaves at three maturity stages.

| Variable | Name of Metabolite | VIP Score |
|:---:|:---:|:---:|
| 18 | Methyl palmitate | 1.87 |
| 33 | 7,9-Di-tert-butyl-1-oxaspiro(4,5)deca-6,9-diene-2,8-dione | 1.58 |
| 4 | Tetradecane | 1.51 |
| 3 | Cyclotetradecane | 1.39 |
| 27 | Hexadecanol | 1.38 |
| 1 | Decane | 1.33 |
| 19 | Methylox | 1.22 |
| 11 | 2-Butyl-1-decene | 1.21 |
| 29 | Octadecanol | 1.21 |
| 17 | 5(2,4-Di-tert-butylphenoxy)-5-oxopentanoic acid | 1.20 |

**Table 7.** *Cont.*

| Variable | Name of Metabolite | VIP Score |
|:---:|:---:|:---:|
| 15 | Methyl benzoate | 1.14 |
| 10 | 9-Methyl-1-undecene | 1.09 |
| 34 | Pinostrobin chalcone | 1.07 |
| 25 | Decanol | 1.05 |
| 32 | 6,10,14-Trimethyl-2-pentadecanone | 1.04 |
| 24 | 4-Methylhexanol | 1.00 |
| 12 | (E)-9-Eicosene | 0.96 |
| 16 | Bis(2-ethylhexyl) carbonate | 0.95 |
| 20 | Dodecanol | 0.94 |
| 22 | Stearic acid, butyl ester | 0.94 |
| 36 | Demethoxymatteucinol | 0.94 |
| 7 | 6-Ethyl-2-methyldecane | 0.94 |
| 8 | 2,6,11,15-Tetramethylhexadecane | 0.89 |
| 28 | Intermedeol | 0.86 |

Partial Least Square (PLS)

Variation of metabolites between the leaves of 'Giant Green' at different maturity stages and their correlation with bioactivities (antioxidant and alpha-glucosidase inhibitory activities) was evaluated using PLS analysis. The details about these bioactivities were reported in our previous study [18]. As seen in the biplot (Figure 9C), the young leaves were located at the upper right-hand quadrant on the t1 axis which is influencing a strong relationship with antioxidant and alpha-glucosidase activities compared to mature and old leaves samples. Besides, it also revealed that this PLS model was good. It had the values of $Q^2$ cumulative at 0.903, $R^2Y$ cumulative at 1.000, and $R^2X$ cumulative at 0.980. The metabolites' correlations with bioactivities in leaf samples were deduced from the loading plot; 17 metabolites were identified that contributed to this relationship. These metabolites were 18, 4, 1, 3, 33, 27, 11, 15, 29, 10, 36, 19, 17, 25, 12, 7, and 24. The variable importance to projections (VIP) was further examined to validate the significance variable. This result also aligned with data from VIP coefficients where the metabolite (18) (rt: 23.85) had the highest VIP value than other peaks which indicates it is the highest influencer to antioxidant and alpha-glucosidase activities. The details about the metabolites are summarized in Table 8.

**Table 8.** Summary of strong loading variables that correspond to the partial-least square analysis (PLS) of the 'Giant Green' cultivar of *S. samarangense* leaves at three maturity stages.

| Variable | Name of Metabolite | VIP Score | PCs |
|:---:|:---:|:---:|:---:|
| 18 | Methyl palmitate | 1.89 | PC1 |
| 4 | Tetradecane | 1.65 | PC1 |
| 1 | Decane | 1.50 | PC1 |
| 3 | Cyclotetradecane | 1.40 | PC1 |
| 33 | 7,9-Di-tert-butyl-1-oxaspiro(4,5)deca-6,9-diene-2,8-dione | 1.39 | PC1 |
| 27 | Hexadecanol | 1.38 | PC1 |
| 11 | 2-Butyl-1-decene | 1.27 | PC1 |
| 15 | Benzoic acid, methyl ester | 1.24 | PC1 |
| 29 | Octadecanol | 1.23 | PC1 |
| 10 | 9-Methyl-1-undecene | 1.20 | PC1 |
| 36 | Demethoxymatteucinol | 1.16 | PC1 |
| 19 | Methylox | 1.14 | PC1 |
| 17 | 5(2,4-Di-tert-butylphenoxy)-5-oxopentanoic acid | 1.09 | PC1 |
| 25 | Decanol | 1.07 | PC1 |
| 12 | (E)-9-Eicosene | 1.04 | PC1 |
| 7 | 6-Ethyl-2-methyldecane | 0.99 | PC1 |
| 24 | 4-Methylhexanol | 0.85 | PC1 |

## 4. Discussion

In the present study, the fingerprints of old, mature, and young leaves were found to be similar but slightly different in peak intensity and this might be due to the presence of

the same type or quantity of metabolites. The similarities between the absorbance peaks also are related to the insignificance among leaf stages in antioxidant and antibacterial activities reported in our previous study. The peak at 1710 cm$^{-1}$ which is assigned to the C=O in the phenolic group is believed to have contributed to these bioactivities. Besides, this is in agreement with the finding by Easmin et al. [24], where they found that FTIR spectra for ethanol and water extracts of *Phaleria macrocarpa* fruit look similar because of the similarity in their chemical composition. Besides, the variation of peak detected between the leaf samples also might be related to different levels of enzyme activities for each maturity stage [21]. In addition, Kharbach et al. [29] reported that the resulting compound fingerprint is mostly related to plant maturity, variation of season, and location of geographic. However, the comparison among spectra only cannot provide the final conclusion about the specific fingerprint that contributed to variations between 'Giant Green' leaves at three maturity stages. So, the data of FTIR was further analyzed and subjected to chemometric analysis.

Principal Component Analysis (PCA) is unsupervised multivariate data analysis (MVDA) that is used to reduce the dimensional large dataset and at the same time has preserved important information. The most important information from the dataset is explained in PC1 and the second most important information is explained in PC2. The score plot was used to differentiate among the samples and the loading plot was used to determine the variable contributed to the samples cluster. In the present study, the PCA was established to find the relationship between 'Giant Green' leaves at three maturity stages and identify the functional group that contributed to the sample separation. The PCA result showed that the young leaves were clearly separated from mature and old leaves. But one biological replicate from mature leaves (ML1) is located near the old leaf samples. This might be why some metabolites in ML1 are also present in old leaf samples or their quantity is almost similar. Some of the previous studies also found no defined cluster between *Ficus deltodeia* syconia varieties [30], cabbage cultivar [31], and *Ipomoea aquatica* [32] because of the identicalness of their chemical contents. Besides, *Eugenia uniflora* leaves showed a clear distance of cluster between the different fruit color biotypes due to their distinctive volatile compounds [33].

Unsupervised Hierarchical Cluster Analysis (HCA) is complementary to PCA analysis. HCA was applied to determine the similarities and dissimilarities between the individual experimental samples. The sample with similar in investigated variable matched in the same cluster but the sample showed the highest dissimilarity was arranged in other clusters. The position of the cluster in the dendrogram also takes into account the far position among clusters that shows the highest dissimilarity between the individual samples [34]. The results obtained from this study demonstrate that the leaf extracts at three maturity stages were arranged in three different clusters and might be influenced by metabolite biosynthesis. Lee et al. [35] reported that metabolite presence varies at the young, mature, and old stages of *S. samarangense* cv. pink leaves. The HCA model also revealed that most leaf extracts present in the same maturity stages formed a homogeneous cluster. It was expected that the samples with the same maturity stage was similar because they consisted of metabolites of the same quality and quantity. However, one of the mature leaves (ML1) deviated away from other mature leaf samples, but arranged in cluster 1 belonging to old leaf samples, indicating that the metabolite can also develop differences among leaves at the same maturity stage. This phenomenon also might be affected by environmental factors, cultivar practices, plant ages, and soil factors of wax apple cultivar. The environmental factors such as temperature, light intensities, and climatic change influence the changes of metabolites in plants [36].

Partial Least Square-Discriminant Analysis (PLS-DA) is a supervised multivariate data analysis (MVDA) tool that is gaining more interest, especially in the analysis of metabolomics data. PLS-DA has the capability to improve the classification of experimental data that cannot be achieved by using PCA. Unlike PCA, PLS-DA is focused on class reductive in achieving the separation between the samples. In this study, PLS-DA is

performed to discriminate the 'Giant Green' leaves at three maturity stages based on their FTIR dataset. In good accord with PCA and HCA analysis, one of the samples from the mature leaf stage that is ML1 cannot clearly be separated from the samples of the old leaf stage might be due to the similarities of metabolites in both samples. However, the other samples from young, mature, and old leaves were well improved in their separation than in PCA. The prediction ability of this PLS-DA model in the classification of the leaf samples with different maturity stages has been validated by the achievement of a 93.33% score in the confusion matrix. The maximum data among class (maturity stage) were collected in PLS-DA analysis, making the variable that discriminates in this model may be unlike those with PCA. Most of the discriminative variables detected in FTIR spectra contributed to the classification of young, mature, and old leaves according to variable importance to projection (VIP) coefficients. The highest VIP score represented the stronger variable attributed to the clustering of the sample. The peaks at 2857, 2928, 1709, and 924 cm$^{-1}$ were mostly related to describing the differences between the maturity stages of leaf samples. The previous literature reported that the asymmetrical (2970 cm$^{-1}$) and symmetrical (2856 cm$^{-1}$) of the methylene group ($CH_2$) and C=H bond (980 cm$^{-1}$) in the FTIR spectrum were commonly related to flavonoid structure [37]. Then, the peak around 1718 cm$^{-1}$ could be attributed to the presence of the ester compound [26]. This finding revealed that the flavonoid and ester compounds could be the largest influencer in separating between maturity stages in leaf samples. So, it can be concluded that the variation of metabolites could be attributed to discrimination between three maturity stages of 'Giant Green' leaves. In good accordance with the previous study, as reported by Lee et al. [35], the chemical compounds such as terpene and terpenoid of wax apple leave cv. pink changes during the maturation stages. Other than that, Gouvinhas et al. [38] reported that the oil from three stages of olive fruit was successfully discriminated by using supervised MVDA. They found that the changes in biochemicals happened with the ripening stages of the olive fruit. Considering the discrimination explained in the PLS-DA model, it is proven that the 'Giant Green' leaves were well-classified according to their maturity stages than using PCA.

Partial Least Square (PLS) belongs to supervised MVDA where the Y-axis represented a dependent variable and X-axis represented an independent variable in the PLS model. PLS is used to find the correlation between the two variables that are generated from the dataset of spectroscopic or chromatographic analysis and bioactivity. The validation and prediction of the goodness of the model are evaluated based on $R^2Y$ (variance explained in predictor variable), $R^2X$ (variance explained in response variable), and $Q^2Y$ (variance predictive of the goodness of fit according to cross-validation). A cross-validated correlation coefficient ($Q^2$) value higher than 0.5 indicates a good PLS model. In the current study, the relationship between FTIR spectra absorbance (wavenumber) with biological activities such as antioxidant (DPPH, NO, and ABTS) and alpha-glucosidase inhibitory activity were investigated. However, the information accessed from FTIR was limited because it just provided a clue about the class of metabolite but the specific metabolite that is responsible for activeness in biological activities is still unknown. The relationship between biological activity (Y-axis) and wavenumber (X-axis) of leaf samples at three maturity stages was illustrated in the bi-plot. Bi-plot was the combination of a score plot and a loading plot. Based on the present results, the bi-plot of leaf extracts revealed that the Y-variables (DPPH, NO, ABTS, and alpha-glucosidase) were located near the sample of the young leaf stage. It revealed that young leaf samples were highly correlated with biological activities. The strongest peaks were obtained at 2857, 2928, and 3300 cm$^{-1}$ which possessed the highest value in the loading plot and VIP score and may be responsible for antioxidant and alpha-glucosidase inhibitory activities of young leaves. The peak at 2857 cm$^{-1}$ and 2928 cm$^{-1}$ may be due to methylene stretching of asymmetrical and symmetrical vibration in methoxyl derivative and aldehyde group, and at 3300 cm$^{-1}$ may be assigned to intermolecular hydrogen bond in alcohol, phenol or carboxylic acid. These peaks showed that the possibility of primary metabolites such as carbohydrates, proteins,

lipids, and polysaccharides, and secondary metabolites such as phenolic acids, flavonoid, terpenes, and terpenoids were present in the leaf sample. In good accordance with a previous study as reported by Christou et al. [39] where they found that the most important peaks in the FTIR spectrum were at the 4000–2500 cm$^{-1}$ which indicates the presence of carbohydrate, protein, lipid, and polysaccharide groups. Saidan et al. [40] also revealed that the sharp peak in the range of 1760–1600 cm$^{-1}$ may be characterized by the presence of flavonoid and terpenoid groups. In addition, the leaves of *S. samarangense* have been reported with an abundance of valuable metabolites such as quercetin, ellagic acid, myricetin, lupeol, sitosterol, triterpenes, betulin, *p*-cymene α-pinene, β-pinene and limonene [41]. These metabolites have been proven to have a significant effect on bioactivities such as antioxidant and alpha-glucosidase inhibitory activities [41–43].

Gas chromatography (GC) is the most intensive instrument used for separation of compounds in a mixture [44]. It becomes the crucial tool in identification of compounds especially in drug discovery or pharmacology and food industry fields. In this study, 37 compounds were detected in three maturity stages of 'Giant Green' leaf extracts. However, only six major compounds were identified and present at the same retention time in all of the leaf extracts. The variation of metabolites in the samples may influent their potency in biological activities. Thus, the strongest antioxidant, antibacterial and alpha-glucosidase activities of 'Giant Green' leaves in our previous study [18] could be related to the greater number of metabolites present in each of the leaf extracts. Some of the metabolites from classes of phenolic, triterpenes, ester, alkane, and carbohydrate have been given more attention by researchers because these metabolites can exhibit various pharmacological activities [45–47]. Previous literature had reported that alkane-based compounds such as tetradecane, hexadecane [48], and nonadecane [49] showed antibacterial and antifungal effects. The presence of metabolites such as methyl benzoate; methyl (9Z,15Z)-9,15-octadecadienoate [50,51], diethylene glycol dibenzoate [52] and 9-Eicosene [53] also have potent antibacterial activity. Other than that, Saleh et al. [46] reported that the metabolites based on fatty acid, organic acid, phenolic acid, carbohydrate, alkane, and sterol may possess alpha-glucosidase inhibitory activity. Fatty acids such as palmitic acid and stearic acid were known to exhibit potent alpha-glucosidase inhibitory activity [54,55] as well as possess strong antioxidant and antibacterial, antitumor, anticholesteremic, immunostimulant properties and anti-inflammatory activities [56,57]. Another metabolite that had the strongest alpha-glucosidase inhibitory activity is phytol [54,58]. Phytol is an acyclic diterpene alcohol and is commonly produced through the degradation process of the plant cell wall [54]. The same metabolite also was reported by other researchers to inhibit the strongest antimicrobial, antioxidant, antinociceptive, and anticancer activities [59,60]. However, the other metabolites found in this analysis might not yet be described in detail by previous literature. Hence, this study revealed that GC-MS is an efficient tool to profile the untargeted peak of the 'Giant Green' cultivar of wax apple leaf samples. However, the huge dataset which was obtained from hundreds of peaks of GC-MS analysis provided a barrier to providing a significant conclusion in terms of specific metabolites that contribute to discrimination between 'Giant Green' leaves at three different maturity stages. Thus, a more manageable size of GC-MS data was obtained by chemometric analysis that applied multivariate data analysis (MVDA).

PCA is performed to reduce the dataset aiming at the structuring of data and clustering of experimental samples. PCA detected the similarities between the samples and classified them into similar clusters. In this study, the 'Giant Green' cultivar of *S. samarangense* leaves did not provide good separation between their maturity stages. The grouping in PCA is based on the strength of variables in the loading plot on PC1 and PC2 axis. Similarly, our findings agreed with the work of Steingass et al. [21]. In their study, one of the green-ripe pineapple fruit did not match with other samples with the same maturity stage and the authors ascribed the variation due to the development of metabolites among the individual fruits that were different even at the same maturity stages. However, our result contradicted a previous study as reported by Maamoun et al. [61] in which there

was clear discrimination between two stages of the ripening stage of *Luffa egyptiaca* Mill fruit. They noted that young fruit exhibits a negative score along PC1 and old mature fruit exhibits a positive score along PC2. Zhang et al. [62] also found a good separation between the three stages of tobacco leaves. The rosette and vigorous growth stages are located along PC1 whereas the mature leaves are located along PC2. The accumulation of certain compounds such as nicotine, sucrose, D-glucose, L-proline, D-fructose, quinic acid, glyceric acid, L-threonic acid, inositol, and DL-malic acid at various quantities in tobacco growth stages were indicated may contribute to this separation. Hence, it can be concluded that the variation and quantity of metabolites in each of the experimental samples played a significant role in the discrimination between them.

Complex chemical reactions occur at each of the maturity stages of plants suitable for their growth and cell development process. So, this process automatically changes the composition of metabolite in the plant. Hierarchical cluster analysis (HCA) is an unsupervised MVDA used to identify the natural grouping between the plant samples characterized by the values of a set of measured properties [63]. The similarity and dissimilarity of the entire set of samples are displayed in the HCA dendrogram. The results revealed that 'Giant Green' leaves were discriminated into three clusters, similar to the results in the PCA. However, each of the clusters did not represent the different maturity stages of leaf samples as expected. It showed that the data as accessed from GC-MS analysis was not able to well-discriminate between 'Giant Green' leaves at three maturity stages. Many factors could be influencing this result such as the similarities of metabolites in each of the samples, location of sampling, and biological replication of samples [30,64,65]. Despite this fact, all of the samples and their biological replicates were collected at a similar location, which has been attributed to the slight differences between metabolites as compared to those samples collected from other locations.

Partial Least Square-Discriminant Analysis (PLS-DA) is further adapted from the unsupervised classification of PCA. The supervised PLS-DA model was applied to investigate the metabolites that contributed to discrimination between 'Giant Green' leaves at three maturity stages. Its results were not in accordance with previous PCA and HCA results where all stages of leaves improved their separation from each other. The validation of the model was also proven with 100% of the confusion matrix result. The metabolites involved in this separation were confirmed with variable importance in the projection (VIP) values. The twenty-four metabolites were identified that consisted of VIP values greater than 0.85 in leaf extracts. From the results, it showed that the variation of metabolites from the groups of alcohol, ester, alkene, alkane, and ketone were involved in the discrimination of leaves (young, mature, and old leaf stages) samples. However, the understanding of factors that influence the discrimination among samples was very complex. Some researchers revealed that the factors of climate, soil, temperature, maturity stage, irrigation, and fertilizer vary the composition of metabolites in plants [31,36,66]. According to Yunusa [30], two possible factors responsible for the separation between the samples are the particular metabolite presence in all samples but different in concentration, and undetected particular metabolites in certain samples. All of these factors also affected the results of PLS-DA analysis. In addition, it was also expected that the PLS-DA model showed better performance in the classification of 'Giant Green' leaves at three maturity stages than PCA since PLS-DA was most effective in discriminating the samples based on their similarities and dissimilarities of metabolite profile.

Partial least square (PLS) is applied to find the correlation between the biological activities (antioxidant and alpha-glucosidase) and metabolites in three maturity stages of 'Giant Green' leaves. From the PLS bi-plot, the young leaves were located near the bioactivities. This finding confirmed the biological activity results, which showed that the sample from young leaf stages had the highest activity compared to samples from other stages. Based on the VIP score, the metabolites contributing the highest to these activities in leaf extract along PC1 were fatty acid (methyl palmitate, 18) and alkane-based compound (tetradecane, 4; decane, 1; cyclotetradecane, 3). This result was consistent with

previous literature showing that the metabolites from fatty acid and alkane derivatives possessed antioxidant and alpha-glucosidase activities [46,54,67]. Anh et al. [68] found that the methyl palmitate presence in *Clausena indica* fruit possesses potent antioxidant and antidiabetic activities. Another study as reported by Murugesu et al. [54], revealed that methyl palmitate and pentadecanoic acid presence in *Clinacanthus nutans* Lindau leaves inhibited the stronger alpha-glucosidase inhibitory activity. The alkanes, fatty acid, methyl ester, and aromatics chain in the essential oil of *Daphne mucronata* Royle leaves and stems showed good antioxidant and antibacterial activities [69]. Considering the relationship as described in the PLS model, it was proven that the metabolites and biological activities in the 'Giant Green' leaves were well-correlated. Then, young leaves also showed the strongest relationship with antioxidant and alpha-glucosidase inhibitory activities than mature and old leaves.

## 5. Conclusions

The ATR-FTIR and GC-MS-based metabolomics approach have well-determined the metabolite variation between the three maturity stages of the 'Giant Green' cultivar of *S. samarangense* leaves. Unsupervised and supervised MVDA were successfully discriminated between the leaf samples and visualized the specific metabolites correlated to the biological activities (antioxidant and alpha-glucosidase). Thus, this work concludes that spectroscopy and chromatography fingerprinting coupled with chemometrics can be applied to select the best maturity of 'Giant Green' leaves for further use in the pharmacological field.

**Author Contributions:** Data curation, N.S.I. and M.M.K.; funding acquisition, M.M.K.; investigation, M.M.K.; methodology, N.S.I., Z.M.R. and M.M.K.; supervision, M.M.K., N.M. and Z.M.R.; validation, A.M., K.M. and M.M.A.; writing—original draft, N.S.I. and M.M.K.; writing—review and editing, N.M., M.M.K., M.M.A. and A.F.M.A. All authors have read and agreed to the published version of the manuscript.

**Funding:** This research was supported by the Ministry of Higher Education, Malaysia (MOHE), project grant No: RACE/F1/SG5/UNISZA/5 with the collaboration of Universiti Sultan Zainal Abidin, Terengganu and Universiti Malaya, Kuala Lumpur, Malaysia.

**Data Availability Statement:** The data related to the findings of this research are available upon request from the corresponding author.

**Acknowledgments:** All of the authors wish to thank the Center for Research Excellence and Incubation Management (CREIM), Universiti Sultan Zainal Abidin, Campus Gong Badak, Terengganu, Malaysia for supporting this project and publication of the findings. The Researchers also acknowledge the Deanship of Scientific Research, Taif University for editing and partial publication support.

**Conflicts of Interest:** The authors declare no conflict of interest.

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
