# Peer review of "Discrimination of Syzygium samarangense cv. ‘Giant Green’ Leaves at Different Maturity Stages by FTIR and GCMS Fingerprinting"

_horticulturae, doi:10.3390/horticulturae9050609_

Round 1

Reviewer 1 Report

This study (horticulturae-2348015) uses FTIR and GC‒MS with chemometric tools to discriminate between different maturity stages of 'Giant Green' Syzygium samarangense leaves, finding a strong correlation between young leaves' metabolite quantities and biological activities.

The manuscript is interesting, topic (introduction, material and methods, results and discussion), can be considered for publication in Horticulturae, specifically the Special Issue "Morphology, Palynology and Phytochemicals of Medicinal Plants", based on the combination of analyses between FTIR and GC‒MS.

Minor points:

L48. carotenes;

L76. Check and confirm scientific name “Curcuma xanthorrhiza Roxb. (family: Zingiberaceae)”, please add italic.

L84-85. Delete. It not necessary.

L123. Why any 530 cm-1?

L179 and 189. Superscript;

L254. These a subtopic?

Figure 5. need correctio, in legends of figures, (A). What is x and y in (B)?

Need descript a VIP score in material and methods, and improve in results sections.

Figure 7 is poor. Not acceptable for publication. Please recreate the image and improve its quality.

Table 7. These a “box” not a table. Recreate.

Combine the figures involving PCA into a single one, as 11 figures are often repetitive. Either merge them or place them in the supplementary material. Similarly, consider revising some of the tables that are not informative but only contain "a value" without substantial explanations and justifications.

In addition, check old references and update when necessary for most recent literature.

Best regards

The manuscript requires a few adjustments in grammar and spelling for better clarity in English.

Author Response

Responses to reviewer:

Horticulturae, MDPI

Manuscript title: Discriminating of Syzygium samarangense cv. ‘Giant Green’ Leaves at Different Maturity Stages by FTIR and GCMS Fingerprinting

Dear Reviewer,

Thank you for your useful comments and suggestions for improving our manuscript. We have made corrections to our manuscript, according to your comments and suggestions. All the corrections are highlighted with blue color on the revised manuscript. These are listed below;

Minor points:

L48. carotenes; L76. Check and confirm scientific name “Curcuma xanthorrhiza Roxb. (family: Zingiberaceae)”, please add italic. L84-85. Delete. It not necessary. L123. Why any 530 cm-1? L179 and 189. Superscript; L254. These a subtopic?

√ We already made corrections above minor mistakes

Figure 5. need correctio, in legends of figures, (A). What is x and y in (B)?

√ Corrected already

Need descript a VIP score in material and methods, and improve in results sections.

√ Already provided which highlighted with blue colour

Figure 7 is poor. Not acceptable for publication. Please recreate the image and improve its quality.

√ We tried to improve the figure but failed to do it. We don’t have better image than that.

Table 7. These a “box” not a table. Recreate.

Combine the figures involving PCA into a single one, as 11 figures are often repetitive. Either merge them or place them in the supplementary material. Similarly, consider revising some of the tables that are not informative but only contain "a value" without substantial explanations and justifications.

In addition, check old references and update when necessary for most recent literature.

√ We already reduce the number of figures and tables according to your suggestions. Old references already replaced by new one. Thank you

The manuscript requires a few adjustments in grammar and spelling for better clarity in English.

√ I already checked the minor mistakes and made corrections already. I think the quality of manuscript is okay, because another two reviewers didn’t raised question about the language quality.

We resubmit the revised manuscript to the journal ‘horticulturae’. We look forward to your positive response.

Thank you & kind regards,

Mohammad Moneruzzaman Khandaker, PhD

Senior Lecturer, Faculty of Bioresources and Food Industry University Sultan Zainal Abidin, Campus Besut, Terengganu, Malaysia

Reviewer 2 Report

Review of horticulturae – 2348015

In the article, Fourier Transform Infrared Spectroscopy (FTIR) and Gas Chromatography-Mass Spectrometry (GCMS) coupled with chemometric tools were applied to discriminate between the different stages of leaves namely young leaves, mature leaves, and old leaves of the Syzygium samarangense cultivar. The chemical variability among samples was evaluated by using Principal Component Analysis (PCA) and Hierarchical Clustering Analysis (HCA) techniques. For discrimination, Partial Least Square Discrimination Analysis (PLS-DA) was applied and then, Partial Least Square (PLS) was used to determine the correlation between biological activities (antioxidant and alpha-glucosidase inhibitory assay) and maturity stages of ‘Giant Green’ leaves. As a result, PCA, HCA and PLS-DA of FTIR and GC-MS data showed the separation cluster between the maturity stages of leaves. PLS results demonstrated that the young leaves show strong correlation between metabolite quantities and biological activities. FTIR and GC-MS coupled with chemometric can be used as a rapid method for discrimination of bioactive structural functions in relation to the biological activity of leaves in different stage of maturity (yang, medium and old). Especially young leaves are rich in bioactive compounds with antioxidant activity and can be used for pharmacological utilization.

To the article, I have next comments and recommendations:

·         In the article, Latin names of plants should be corrected and in italics. E.g., Curcuma xanthorrhiza; Phaleria macrocarpa; also in references: Brassica rapa ssp. pekinensis.

·         L. 79: Soxhlet..

·         In centrifugation convert rpm, e.g., 4000 rpm to g (rcf).

·          In the used equipment should be completed data about producer, town, and country, e.g., in: Shimadzu Prestige-21 Spectrophotometer (Shimadzu Brand, town, country); Deuterated Triglycine Sulphate (DTGS) detector (producer, town, country); Golden 116 Gate Single Reflection Diamond ATR accessory (producer, town, country); GC-MS Agilent (19091S-433UI system) machine (Agilent, town, country).

·         Correct some units with superscript, like cm-1 (L. 118, 201, 202).

·         Table 5, 9 and 10: (E)-9-Eicosene.

·          In the text better:…leaf samples (L. 370, 373, 379, 506, 511) instead „leaves samples“.

·         L. 554:… p-cymene, α-pinene,…

·         In the References some abbreviations of journals titles should be corrected to italics, e.g., Anal.; Bioanal. Chem.; Food Chem.  

·         Reference 33:

Mesquita, P. R. R.; Nunes, Dos Santos, F. N.; Bastos, L. P.; Costa, M. A. P. C.; Rodrigues, F. M.; Andrade, J. B. Discrimination of Eugenia uniflora L. Biotypes based on volatile compounds in leaves using HS-SPME/GC–MS and chemometric analysis. Microchem. J. 2017,130, 79-87.

·         The citation Mesquita et al. (2017) is missing in the text, it must be added.

  In the article, Latin names of plants should be corrected and in italics. E.g., Curcuma xanthorrhiza; Phaleria macrocarpa; also in references: Brassica rapa ssp. pekinensis.

·         L. 554:… p-cymene, α-pinene,…

·         In the References some abbreviations of journals titles should be corrected to italics, e.g., Anal.; Bioanal. Chem.; Food Chem.  

Author Response

Horticulturae, MDPI

Manuscript title: Discriminating of Syzygium samarangense cv. ‘Giant Green’ Leaves at Different Maturity Stages by FTIR and GCMS Fingerprinting

Dear Reviewer,

Thank you for your useful comments and suggestions for improving our manuscript. We have made corrections to our manuscript, according to your comments and suggestions. All the corrections are highlighted with blue color on the revised manuscript. These are listed below;

  • In the article, Latin names of plants should be corrected and in italics. E.g., Curcuma xanthorrhiza; Phaleria macrocarpa; also in references: Brassica rapa ssp. pekinensis.
  • L. 79: Soxhlet. In centrifugation convert rpm, e.g., 4000 rpm to g (rcf).

√ Already corrected above mentioned minor mistakes

  • In the used equipment should be completed data about producer, town, and country, e.g., in: Shimadzu Prestige-21 Spectrophotometer (Shimadzu Brand, town, country); Deuterated Triglycine Sulphate (DTGS) detector (producer, town, country); Golden 116 Gate Single Reflection Diamond ATR accessory (producer, town, country); GC-MS Agilent (19091S-433UI system) machine (Agilent, town, country).

√ Has been corrected.

  • Correct some units with superscript, like cm-1 (L. 118, 201, 202). &Table 5, 9 and 10: (E)-9-Eicosene.

√ Has been corrected.

  • In the text better:…leaf samples (L. 370, 373, 379, 506, 511) instead „leaves samples“. · L. 554:… p-cymene, α-pinene,…
  • In the References some abbreviations of journals titles should be corrected to italics, e.g., Anal.; Bioanal. Chem.; Food Chem.

√ Has been corrected.

  • Reference 33:

Mesquita, P. R. R.; Nunes, Dos Santos, F. N.; Bastos, L. P.; Costa, M. A. P. C.; Rodrigues, F. M.; Andrade, J. B. Discrimination of Eugenia uniflora L. Biotypes based on volatile compounds in leaves using HS-SPME/GC–MS and chemometric analysis. Microchem. J. 2017,130, 79-87.

  • The citation Mesquita et al. (2017) is missing in the text, it must be added.

√ Already added

Comments on the Quality of English Language

  In the article, Latin names of plants should be corrected and in italics. E.g., Curcuma xanthorrhiza; Phaleria macrocarpa; also in references: Brassica rapa ssp. pekinensis.

√ Has been corrected.

  • L. 554:… p-cymene, α-pinene,…
  • In the References some abbreviations of journals titles should be corrected to italics, e.g., Anal.; Bioanal. Chem.; Food Chem.

√ Has been corrected.

We resubmit the revised manuscript to the journal ‘horticulturae’. We look forward to your positive response.

Thank you & kind regards,

Mohammad Moneruzzaman Khandaker, PhD

Senior Lecturer, Faculty of Bioresources and Food Industry University Sultan Zainal Abidin, Campus Besut, Terengganu, Malaysia

Reviewer 3 Report

The manuscript presents a method for discrimination of Syzygium samarangense leaves at different maturity stages, assessing the results of metabolite variability and discriminating according to biological activity. My opinion is that the manuscript is publishable in its current form.

Author Response

Horticulturae, MDPI

Manuscript title: Discriminating of Syzygium samarangense cv. ‘Giant Green’ Leaves at Different Maturity Stages by FTIR and GCMS Fingerprinting

Dear Reviewer,

Thank you for your valuable comments on our manuscript.

The manuscript presents a method for discrimination of Syzygium samarangense leaves at different maturity stages, assessing the results of metabolite variability and discriminating according to biological activity. My opinion is that the manuscript is publishable in its current form.

√ Thank you for the comments.

We resubmit the revised manuscript to the journal ‘horticulturae’. We look forward to your positive response.

Thank you & kind regards,

Mohammad Moneruzzaman Khandaker, PhD

Senior Lecturer, Faculty of Bioresources and Food Industry University Sultan Zainal Abidin, Campus Besut, Terengganu, Malaysia
